# Separation of Leather, Synthetic Leather and Polymers Using Handheld Laser-Induced Breakdown Spectroscopy

**DOI:** 10.3390/s23052648

**Published:** 2023-02-28

**Authors:** Nicole Gilon, Margaux Soyer, Mathilde Redon, Patrice Fauvet

**Affiliations:** 1Institut des Sciences Analytiques, ISA UMR 5280, CNRS, Université Claude Bernard Lyon1, 69622 Villeurbanne, France; 2IDS Foods, 5 Av. Lionel Terray, 69330 Meyzieu, France

**Keywords:** leather, polymers, synthetic leather, LIBS, principal factor analysis

## Abstract

Genuine leather is produced from animal skin by chemical tanning using chemical or vegetable agents, while synthetic leather is a combination of fabric and polymer. The replacement of natural leather by synthetic leather is becoming more difficult to identify. In this work, Laser Induced Breakdown Spectroscopy (LIBS) is evaluated to separate between very similar materials: leather, synthetic leather, and polymers. LIBS is now widely employed to provide a specific fingerprint from the different materials. Animal leathers processed with vegetable, chromium, or titanium tanning were analyzed together with polymers and synthetic leather from different origins. The spectra exhibited typical signatures from the tanning agents (Cr, Ti, Al) and the dyes and pigments, but also from polymer characteristic bands. The principal factor analysis allowed to discriminate between four main groups of samples representing the tanning processes and the polymer or synthetic leather character.

## 1. Introduction

Sorting material using Laser Induced Breakdown Spectroscopy (LIBS) has received much attention in the past years. Recycling and on-line control requires fast and sensitive techniques likely to be introduced directly on conveyors and production sites, so that on-line or portable devices are more frequently employed. The metal industry (steel or aluminum) requires more separation techniques that are likely to sort refractories and metal scrap in order to recycle used components and reinject them in the production of new metals [1]. LIBS has gained much attention due to its distant and fast analysis and also, in combination with advanced chemometric tools, due to its ability to identify a material out of the rich LIBS spectral information. In the field of food control, the issue of traceability is often addressed using LIBS spectroscopy as the elemental composition of a vegetable reflects the soil and, therefore, the geographic origin of the product. In the case of coffee, wine, or tea, LIBS has shown a good potential to sort products according to their geographic origin [2,3,4]. Detecting adulteration of food products has been also investigated for milk products, coffees, meats, and olive oils [2,5,6]. Development of the traceability activity has led to a rich literature about the tracers that are likely to identify the product origin. Likewise, the research conducted on sorting polymers in view of either recycling activities or remediation is essential to remove potentially toxic plastics from the recycling processes when analyzing used plastics or waste electrical and electronic equipment (WEEE) [7]. The tracers use either elemental or molecular information, as the LIBS spectra contains specific molecular bands such as C2, CN, or NH [8]. The identification of the authentic origin or a specific material against altered or synthetic samples is also important in the leather industry, and some research is examining this subject using LIBS analysis [9,10]. Synthetic leather consists of a textile with an application of polymer in the form of a lacquer or an emulsion. Most common polymers employed are polyurethane, polyvinyl chloride, or polypropylene [11]. The quality evaluation of leathers is essential to ensure their properties, origin, and, of course, to certify their nature, i.e., from animal skins or from artificial sources. Nowadays, synthetic material exhibit an appearance and physical characteristics quite similar to those of natural leather. For instance, the percentage of elongation and the stitch tear strength of synthetic leathers are even better than natural leather [12]. It is therefore essential to be able to distinguish between these very similar materials. The quality evaluation of leathers is fundamental to ensuring their good properties and lack of toxicity. 

In the tanning processes, toxic elements can be incorporated into the leather and the color agents, and tanning agents are mainly composed of metals (Cr, Ti, Al …) which are introduced at high concentrations. During subsequent coloring and finishing processes, the skins are treated with pigments and dyes also containing heavy metals (Fe, Ni …) [13].The discrimination between synthetic and natural leather, as well as the identification of the employed tanning process, is a way to evaluate if the skin is of the highest quality or not. Some waste leather skins may be bonded together, using polyurethane, onto a fiber sheet. 

In addition to physical testing of leather and synthetic leather, many classical laboratory methods are employed to identify and quantify elements in natural leather. These include ICP-MS (Inductively Coupled Plasma Mass Spectroscopy), ICP-OES (Inductively Coupled Plasma Optical Emission Spectroscopy), or MP-OES (Microwave Induced Optical Emission Spectroscopy) using a prior digestion step which requires to destroying a large part of the sample but leads to very high sensitivity [14,15]. The XRF (XRay Fluorescence) analysis provides a real, non-destructive method, and is therefore very attractive [11,16] but also less sensitive. The specific INAA (Instrumental Neutron Activation Analysis) technique is very sensitive and non-destructive, but is quite complex to employ, as was described in different papers [17] due to the interest in leather and soil analysis without preparation, as the tanning industry is frequently suspected of polluting soils in the environment. When dealing with leather analysis, the concentration of tanning agents is high enough to employ less sensitive atomic spectroscopic methods such as flame atomic absorption spectrometry [18]. The levels of Cr or Ti are several% *w*/*w*, and the concentration of Na, Mg, K, or Ca employed as additives to modify pH during the tanning process also reaches several%. Furthermore, the molecular composition and, especially, the structural variation of tanned skin is analyzed using vibrational spectroscopy techniques such as Raman spectroscopy and Attenuated Total Reflectance–Fourier Transform InfraRed (ATR-FTIR). In this view, specific band analysis and chemometric methods are used to identify variations which affect the physical properties of leather [19]. Raman microscopy gives information on the chemical structure and allows to identify organic coloring agents [20]. Fourier-Transform Infrared (FTIR) spectroscopy has been known as one method to identify the origin of material based on the identification of a series of molecules typical of the skin or leather sample. For instance, the lipid profile is employed to discriminate between pig and goat leather using lipid extraction and further FTIR analysis [21]. The surface of the leather is also investigated using ATR-FTIR spectroscopy. The leather surface is treated using different components: resins, oils, waxes, or resin treatments, and their identification is useful to inform consumers [22]. 

In view of an efficient discrimination of materials by LIBS, the different emissions, including elemental and molecular information, may be combined. This complex spectral information and the need to make it clear has led to a rich chemometric literature. As for most infra-red or Raman studies, this combination has been used for the discrimination of different types of samples, such as polymers employed in toys, common polymers, or pigments, using Principal Component Analysis (PCA), K nearest neighbors (KNN), Soft Independent Modeling of Class Analogy (SIMCA), and Partial Least Square for discriminant analysis, PLS-DA [23,24,25]. The LIBS fingerprint has also been applied to very similar sample such as meats or hair. In the case of meat, an adulteration of meat using offal could be identified using selected line measurements on the LIBS spectra and further PCA analysis [5]. If the elemental fingerprint is typical of the sample nature, a lack or a deficiency in the main element compared with the average level may reflect a problem. From this assumption, the concentration of elements in nails has been employed to distinguish between healthy and alcoholic or opium addicted patients [26]. In several cases, the major element concentrations are affected by the addiction or illness [27]. Chemometrics are also employed for calibration purpose; PLS and even PCR models are shown to provide robustness and precision for metal determination in polymers; RF (random forest) with spectra calibration is also efficient for multielement calibration [28,29]. To examine the data, an unsupervised method is frequently employed in order to reduce the data obtained either from the complete spectra (i.e., more than 20,000 pts for each spectra) or from a selected list of measured wavelengths that represent a limited number of elements (10 to more than 50) [5,9]. In the case of complete spectra, the score plots often represent the model spectra that allows to discriminate samples. When naturally occurring classes exist, most studies use supervised tools such as discriminant function analysis (DFA), which further allows to determine which variables discriminate between the known naturally occurring groups [26]. Once the model is well validated, prediction is employed for unknown samples. Such operations are performed using KNN, Soft Independent Modeling of Class Analogy, SIMCA, or PLS-DA [30]. One of the key feature of LIBS measurements is the combined molecular and elemental data obtained from the same spectrum. The advantage over other elemental analysis methods is that it allows, when combined, a multivariate analysis to discriminate between very similar samples such as polymers, synthetics, and genuine leather [7,24].

The objective of this work was to identify and highlight very subtle variations in the LIBS spectra of synthetic leather, genuine leather, and polymers (as these are employed to produce synthetic leather) combined with chemometric methods. For this purpose, LIBS spectral data from 104 samples, including 34 polymers, 60 leather, and 10 synthetic leather data were acquired. A new factor analysis approach allowed pinpointing the elements and molecular fingerprints likely to discriminate the samples.

## 2. Materials and Methods

### 2.1. Z-300 Handheld LIBS Analyzer

The work reported here utilized a Z-300 LIBS analyzer (Quantum RX, Saint-Aubin, France); it is a handheld instrument with dimensions of 21 × 29 × 11 cm that weighs 1.8 kg. The laser is a PULSAR^TM^ (Nd:YAG 1064 nm diode-pumped solid-state pulsed laser), producing a 100 μm focused beam that delivers a 5–6 mJ pulse to the sample with a 1 ns pulse duration. The employed repetition rate was 50 Hz. The analyzer operates with argon purging directly to the focusing area on the sample surface where LIBS plasma formation occurs, thus producing plasma containment and emission signal enhancement.

The LIBS emission is collected and injected into three internal spectrometers utilizing time-gated CCD detectors. Z-300 spectrometer records a broad range of plasma light emissions, from 190 to 950 nm. The respective spectral ranges and resolutions are: 190–365 nm with FWHM of 0.18 nm, 365–620 nm with FWHM of 0.24 nm, and 620–950 nm with FWHM of 0.35 nm. The delay time between the laser pulse and signal integration is variable between 250 ns and several µs. The integration time is also variable in the millisecond range.

Signal measurements were obtained from grids of spots, typically 12 spots covering a 2 × 2 mm surface. This grid is repeated on five different locations on the target surface to ensure representative measurement.

All acquired spectra were converted into an Excel file treated by xlstat software (Addinsoft (2022)-XLSTAT New York, NY, USA). The normalization procedure is similar to that employed in previous work [4].

### 2.2. Samples

The leather samples were bought from local professional providers; the plastics were from a previous project [24]. Samples consisted of 56 pieces of animal skin leather: sheep, ostrich, reptile leather (snake, crocodile), and fish leather (stingray skin, called “Galuchat”), 10 artificial leathers, and 34 plastic chips (3 cm diameter). The color and nature of the samples are detailed in Table 1.

LIBS spectra were collected from a 12 × 6 grid with 8 shots at the same position; each spectrum was therefore the average of 576 shots. Repetition of measurement at the sample surface re-enforced the consistency of the collected data and partially compensated for the sample inhomogeneity.

## 3. Results

### 3.1. Signal Relevant for Fingerprint Identification

The basic principle behind fingerprinting leather, synthetic leather, and polymer is that the sample composition (i.e., trace element, molecular signals, etc.) will reflect their nature. For example, the large occurrence of titanium or chromium suggests the sample is tanned, i.e., it could be a leather. However, in synthetic products and also in polymers, both titanium and chromium may also originate from color pigments or polymer treatment. 

The leather treatment from chromium or titanium produces very intense lines due to the over-percent concentration of the element contrasting with the low intensity spectra of other leather (Figure 1). As seen from Figure 1 and Figure 2, the spectral fingerprint of the titanium tanning process produces a complex spectrum. From our sample set, many different animal skins were found to be titanium tanned (38%), while chromium tanning was 40% on a total data set of 56 leather samples. Vegetable tanning represents 20% of our sample set. Some leather contained both chromium and titanium, probably due to the color pigments employed. Leather spectral fingerprints (Figure 1) are also characterized by the unhearing protocol; animal skin is treated with chemical agents such as sodium hydroxide and calcium hydrosulfide. Alkalinity removal requires acids. Nevertheless, the ions from alkalis remain, and strong signals of both Na and Ca appear on all the leather spectra. The further tanning process requires solutions with high concentrations of tanning agents, metal sulfates, such as Cr, Al, and Ti, and vegetable tannins. As can be seen from Figure 1, if Ti and Cr tanning are producing high intensities for the specific lines of both elements, when no Ti or Cr is employed, the signals from Na and Ca are even more intense. In finishing operations, pigments, solvents, oils, colorants, and coating products are added to give shine and softness to the leather. Furthermore, the animal skin itself brings elemental characteristics of the animal. It can be observed that leather from fish origin (stingray) contains a very high amount of phosphorus, as seen from the occurrence of 213.62 nm and 214.92 nm phosphorus lines. Elements from pigment dyes are also relatively intense, such as iron or copper, as most of the brown colored leathers are tinted using iron oxides. The iron level is different from light brown or “gold” color to dark brown; as seen from spectra in Figure 2, the Fe 438.38 nm line intensity increases as the color gets darker. 

Furthermore, some of the so-called vegetable tanned leather also exhibited brown colors, with an important iron content that also produces a complex spectrum. Nevertheless, the addition of sodium, calcium, aluminum, or magnesium to leather gave strong signals (Figure 3). 

In contrast, polymer exhibited low intensity spectra with only molecular bands and low elemental information provided only by the additives, such as Zn, Ca, P, or Sb, which are often employed as fillers, flame-retardants, or UV protective agents.

### 3.2. LIBS Parameters Optimization

The selection of the tracers specific to each type of material (polymer, leather, and synthetic leather) relies also on the careful selection of LIBS parameters to obtain maximum information and reliable data from the samples. Since the spectra of most samples are complex, as seen from previous discussion, the selection of parameters is crucial. To highlight the difference between polymers and leathers, the sensitivity optimization should improve the molecular information, and the selection of data should not be hindered by titanium or iron interferences. The LIBS device was used in the gated mode, and integration time and delay were evaluated. The delay and integration times available on the Z300 handheld instrument typically range from 250 ns to 10 µs for the delay and from 1 ms to 18 ms for the integration time. A study of the element atomic line (P 214.92 nm), molecular bands (CN 388.34 nm, C2 516.52 nm), background intensity, and noise (RSDb relative standard deviation (%) of the background at 400.00 nm) is presented in Figure 4. Measurements are shown from polymer analysis. Using a short delay, an increase in integration time was evaluated (Figure 4, left), then, using a fixed integration time (1 ms), the delay was tested from 250 ns to 20 µs (Figure 4 left). It was verified on a smaller number of measurement that a similar trend is obtained for leather: stingray for the phosphorus line, snake and stingray for the CN and C2 molecular bands.

The delay is critical for detection of elemental signatures, with a 10 to 20% drop in the signal by only increasing the delay to 500 ns, so that a short delay was kept (250 ns). The integration time allows keeping sensitivity for values in the 1 to 6 ms. However, a shorter value gave the best results. The signal drop is explained by the rapid cooling of the plasma, which reduces line excitation. The background itself is reduced with an increased integration time, but the noise increase is dramatic. Accordingly, a short value of 1 ms was defined as a good compromise. Using the optimized delay and integration times (250 ns and 1 ms, respectively), the high laser repetition rate (50 Hz) made it possible to average the signal over hundreds of accumulated spectra with minimum analysis time.

### 3.3. Study of Molecular LIBS Information

Leather is a protein-based material, containing natural polymers such as collagen or keratin. These proteins are polyamides, and are, in fact, very similar to nylon. A typical spectrum of each sample type is presented in Figure 5a,b; the complete ranges of 360–390 nm and 500–520 nm are presented, and the relevant lines and molecular bands are identified. As the molecular signals from C2, CN, CH, O, H, are likely to distinguish between polymer types [7], a first unsupervised principal component analysis (PCA) plot using the spectral intervals containing the molecular information is employed. The preprocessing of the dataset involved background correction and normalization by the total spectra intensity as it reflects the small variations due to shot-to-shot instrumental variations.

As presented from Figure 6a, molecular information provided by the C, H, O, and N lines, as well as the C2 and CN bands, and the common ratios usually employed for polymer discrimination [30] allows to partially distinguish between polymers and leather samples; blue circles representing leather are low on the PC1 axis (<0), while polymers are high in this axis. Components PC1 and PC2 explained 58% of the cumulated variability. The first PC captured 35.6% of the variance, the second PC captured 24.4% of the variance, and the third PC captured 16.4%. Using the three first components, more than 75% of the variability is explained (Figure 6b). The score plot (not presented) showed that leather is characterized by the occurrence of intense signals from N, H, and O, whereas polymers exhibited more intense signals on the wavelengths corresponding to C2 and C. PC2 was characterized mainly by the CN signal; this signal is often present in all samples. As a result, PC2 did not separate between polymer and leather.

Data reduction from 11 measured lines to only 3 components is important. However, the synthetic leathers (yellow circles from Figure 6a) and many of the colored leather samples are located between the main two groups with no clear separation. 

An examination of the molecular signatures (Figure 5) shows that the molecular signals on the LIBS spectra are very limited for some leathers. It is observed that the organic fingerprint provided by the CN and C2 bands is limited on the titanium-tanned leather, as seen from spectra of the cow leather treated with titanium (blue spectra from Figure 5a,b). The C2 bands from many leathers appeared very low whereas, the CN band could be distinguished. These typical molecular signatures appeared clearly on crocodile leather and polymers. The polymers all exhibited intense C2 molecular bands, and when CN is present in the material, the CN bands are also clearly distinguished. Their spectra is most of the time dominated by these strong signals. When the leather are examined, their surface treatment, performed either to obtain an aniline or semi-aniline finish, may contain chemicals likely to increase the intensity of both C2 and CN. Depending on the treatment, the polymeric character is reinforced or not.

### 3.4. Combination of Molecular and Elemental Information

The data provided by the complete spectra are often employed in LIBS spectroscopy to discriminate between samples. In the case of leather, the complex fingerprint caused by titanium occurrence in many leathers could bias the discrimination with PC only based on the Ti information. A supervised selection of elements and wavelengths minimizes this risk. The 44 lines and molecular bands selected according to their sensitivity and resolution from other lines are presented in Table 2. Many elements were detected in leather, probably due to the pigments employed to obtain very bright colors, which are often applied for fish leather or snake leather (purple, yellow, bright orange). Large additions of calcium, magnesium, sodium, or barium salts are also visible from yellow- and turquoise-colored snake and crocodile skins.

A principal component factor analysis (using a Varimax rotation to determine the factor loadings) was exploited to determine the common features responsible for the correlation structure among the variables: elemental (lines) and molecular information. Factor analysis allows to identify a hidden variable, assuming that correlation between measured variables can be explained by this common factor. Taking into account the 50 lines for each leather or polymer, the factor analysis was performed and is presented in Figure 7.

The 3D representation of the factor analysis is shown in Figure 7. Cluster analysis shows that the leathers are well separated in two distinct groups (LowT and HighR). The polymers (POLY) are also discriminated from the first two groups and from the synthetic-leather (SIM) samples that appear dispersed between the clusters. The most important group is POLY; the polymers are very similar in comparison with leathers so that the 34 samples appeared as a very compact cluster. Three leather samples appeared also as outliers (orange balls) with distinct values for the first three factors. The three leathers contained both Cr and Ti together with a dark brown color so that they could belong to the two leather groups.

The first leather group (LowT) is characterized by negative F1 and F2 values and F3 close to zero. The second leather group (HighR) is defined by a high F1 and a low F2. The factor patterns F1, 2, and 3 are plotted in Figure 8. The first three factors represented 56.6% of the total variance. F1 captured 28.8% of the variance. Axis F1 represents the complexity of the spectra as it is defined with high positives scores for both Ti-tanned but also Fe-containing leathers with no Ti signal (HighL). This factor reflects the occurrence of a large number of elements but also an important concentration of elements with a rich spectra such as Ti. The F1 factor is also negatively correlated to organic signatures such as the C2 lines intensity, C 193.09 nm, and C 247.85 nm (Figure 8). As observed from the spectra in Figure 5a,b, the large intensity of several elements originating from leather treatment drastically reduces the polymeric aspect of the leather matter. The CN/C ratio is also important in this factor. The second group discriminated by the F1 factor also represents rich spectra from synthetic leather (SIM). 

According to the factor pattern of F2, F2 is positively correlated with the molecular ratios C2/C ratio but also with the C2 intensity, both the C2 512 nm and C2 471 nm molecular bands and H/O. This factor mainly contains the polymeric aspect of the sample. As expected, the polymers are well separated from leathers along axis F2, as seen from the 3D graph. Polymers and leather group B are also well separated in F2. The F2 factor was negatively correlated with line intensity from Si, Fe, Na, and K. The leather LowT group and HighR groups are separated in F3. The three leather groups are also well separated in F3 from synthetics and polymers having negative F3, whereas the leather had a positive F3. The F3 reflects a high positive correlation with Cr, Na, K, and CN, and negative values for C2/C and H/O ratios.

The factor analysis could correctly isolate 90% of the synthetic leather; only one sample is misidentified as a polymer and three sheep leathers appeared as outliers. The score for polymer identification is 98%, and, with regard to the leather groups, LowT contained both the Cr and vegetable tanning samples as no special weigh was given to the Cr intensity. The exotic leathers are present in this group; the levels of Cr or other elements were lower in these samples. The Ti-tanned and iron-containing leathers were present in the HighR group. As mentioned in the discussion, the factor analysis revealed a discrimination from the complexity of the spectra and not on the nature of the tanning process. Further study using a wider selection of vegetable tanned leathers should be performed in order to improve separation using the LIBS spectral information. 

## 4. Conclusions

In this study, a new method was employed for the separation of leather, synthetic leather, and polymer according to their LIBS spectra. Factor analysis is a convenient way to identify the main information responsible for the clustering of studied samples. The influence of the tanning process is clearly important and is clearly identified using the LIBS spectra. It was demonstrated that separation of synthetic leather from other samples was possible. Factor analysis was sensitive to the complexity of the spectra produced when high amounts of pigments or tanning agents are employed. The molecular information and ratios were also efficient at identifying and discriminating between polymers or synthetic leather from other samples.

Compared to other analytical methods, the developed LIBS-based method enabled a fast analysis, and the method is easy to employ on-site as the LIBS instrument is a handheld one. Despite the lower sensitivity compared to laboratory tools, the use of specific fingerprints associated with leather tanning, coloring processes, specific molecular bands, and their ratios was essential for discriminating between polymers and leathers, and, therefore, between leather produced from animal skin and the synthetic leather produced from polymers. 

## Figures and Tables

**Figure 1 sensors-23-02648-f001:**
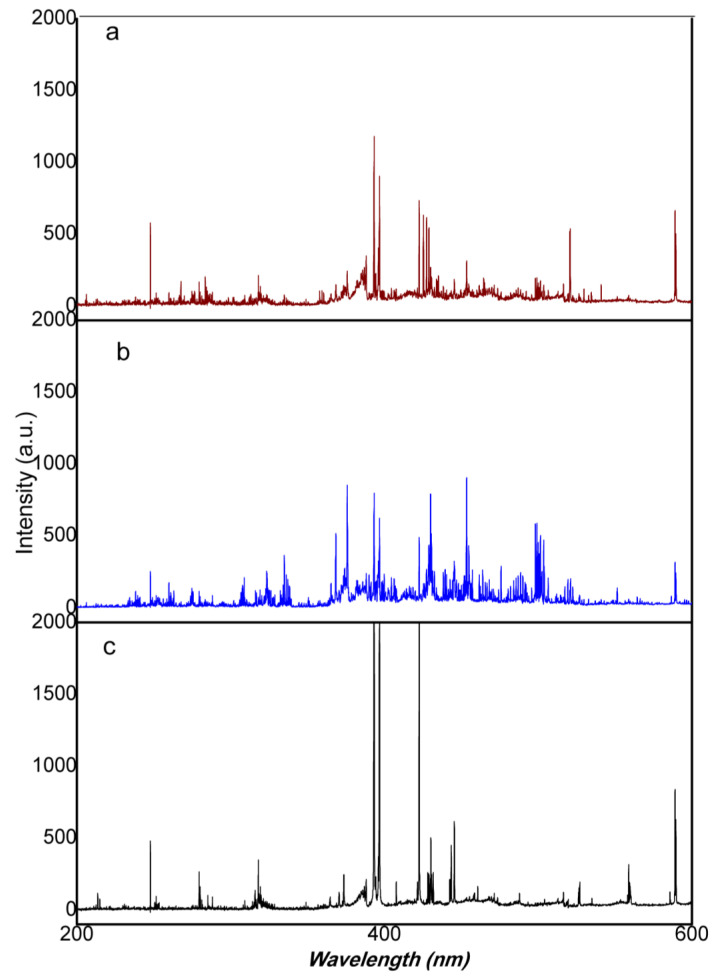
Average spectra of different leather types (**a**) chromium tanning, (**b**) titanium tanning, and (**c**) vegetable tanning.

**Figure 2 sensors-23-02648-f002:**
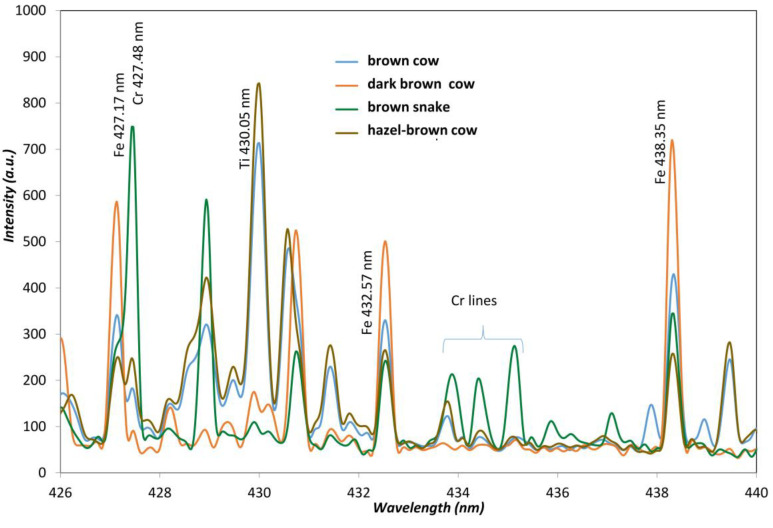
Iron present from brown leathers from cow and snakeskins.

**Figure 3 sensors-23-02648-f003:**
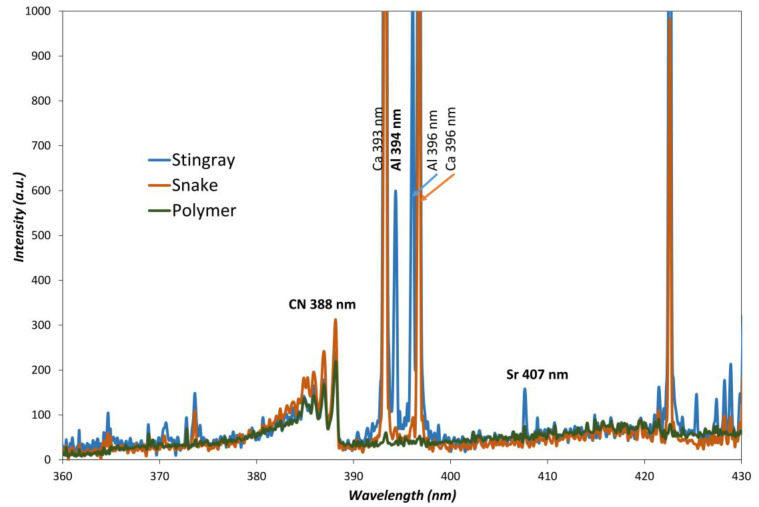
Molecular CN band from polymer (ABS) and different leathers (stingray and snake).

**Figure 4 sensors-23-02648-f004:**
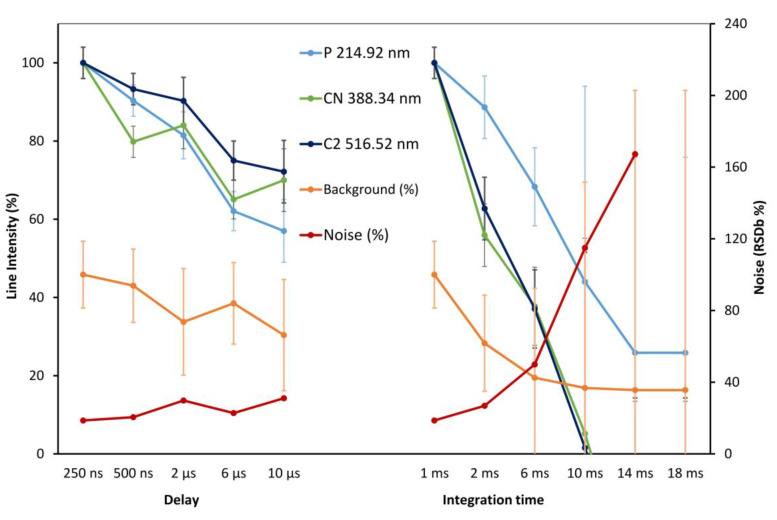
Delay and integration time optimization on the handheld LIBS. Lines and background intensity are normalized to the first measure. Uncertainty is calculated using the five grid replicates on the sample surface.

**Figure 5 sensors-23-02648-f005:**
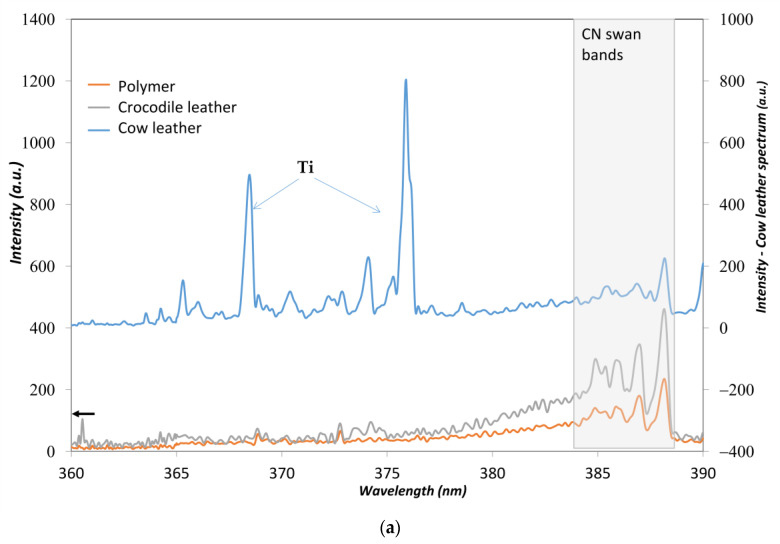
(**a**) CN molecular bands from cow and crocodile leather and polymer ABS. (**b**) C2 molecular bands from cow and crocodile leather and polymer ABS.

**Figure 6 sensors-23-02648-f006:**
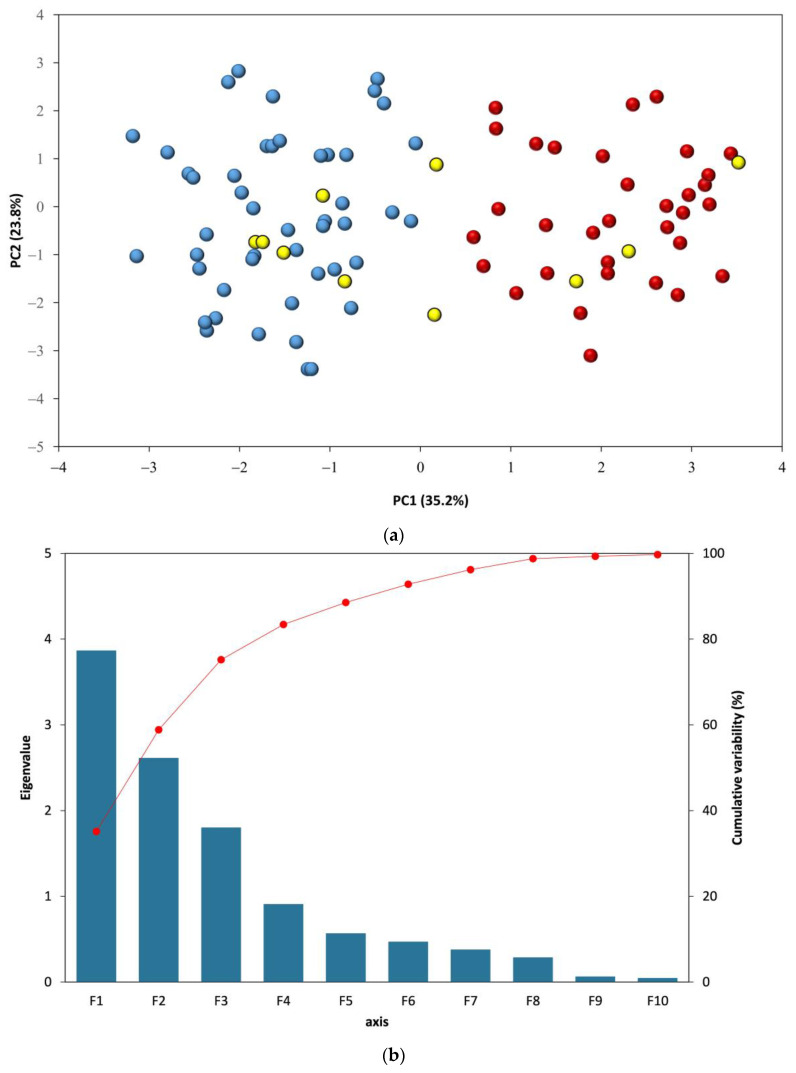
(**a**) PCA realized from the 10 molecular and organic element measurements (blue circles: leather, red circles: polymers, and yellow circles: synthetic leather). (**b**) Scree plot from the PCA shown in (**a**).

**Figure 7 sensors-23-02648-f007:**
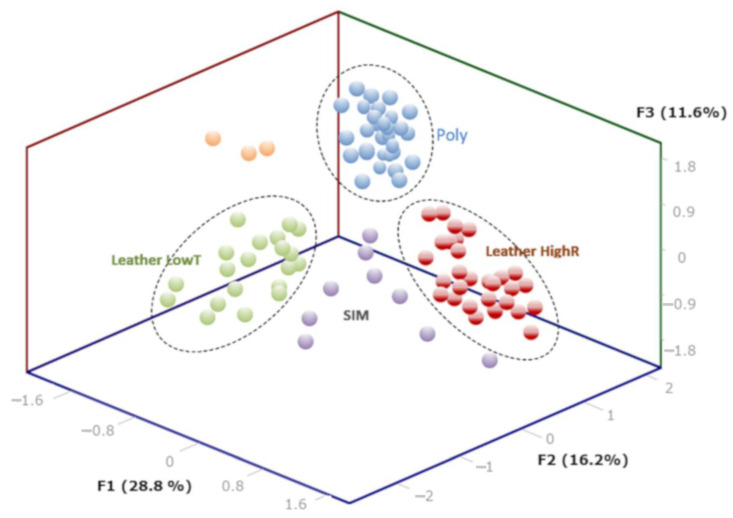
Factor analysis realized from the leather, synthetic leather, and polymers using the lines from Table 2.

**Figure 8 sensors-23-02648-f008:**
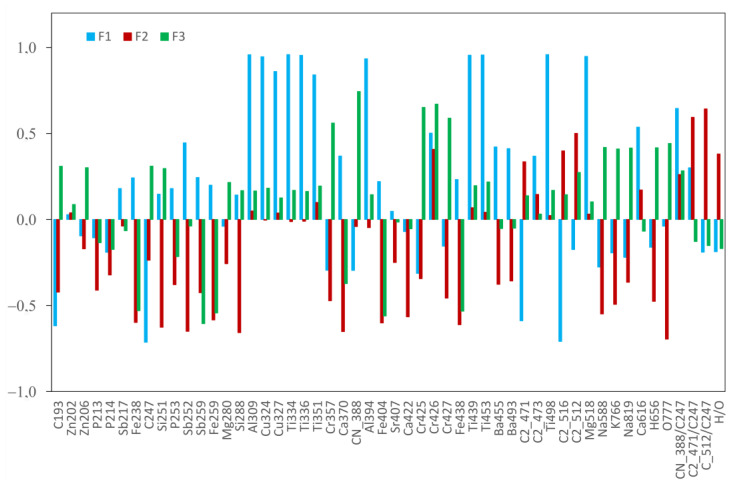
Factor pattern realized from the lines shown on Table 2.

**Table 1 sensors-23-02648-t001:** Samples tested in this work. All samples, including polymers, leather, and synthetic leather had different colors.

Leather	Animal Skin	Number of Pieces
	Cow	16
	Sheep	14
	Ostrich	6
	Crocodile	6
	Buffalo	2
	Stingray	4
	Snake	8
**Synthetic Leather**		10
**Polymer**	**Composition**	
ABS	Acrylonitrile butadiene styrene	8
ABS-PC	Acrylonitrile butadiene styrene and polycarbonate	8
PP	Polypropylene	4
PS	Polystyrene	4
PE	Polyethylene	4
PVDF	Polyvinyl difluoride	4
Rislan	Polyamide	2

**Table 2 sensors-23-02648-t002:** Element wavelengths and molecular bands employed for principal component factor analysis.

Element	λ (nm)	Element	λ (nm)	Element	λ (nm)	Element	λ (nm)
C	193.09	Mg	280.27	Sr	407.70	C2 ^1^	471.52
Zn	202.55	Si	288.15	Ca	422.67	C2 ^1^	473.71
Zn	206.20	Al	309.27	Cr	425.43	C2 ^1^	512.93
P	213.62	Cu	324.75	Cr	426.02	C2 ^1^	516.52
P	214.92	Cu	327.98	Cr	427.48	Mg	518.36
Sb	217.59	Ti	334.03	Ti	439.50	Na	588.90
Fe	238.20	Cr	357.87	Ca	445.50	Ca	616.21
C	247.85	Ca	370.60	Ti	453.32	H	656.28
Si	251.61	Al	394.40	Ba	455.40	K	766.48
P	253.56	CN ^1^	388.34	Cr	464.61	O	777.41
Sb	259.80	Fe	404.12	Ba	493.40	Na	819.48

^1^ Molecular band.

## Data Availability

Not applicable.

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
