# Peer review of "Separation of Leather, Synthetic Leather and Polymers Using Handheld Laser-Induced Breakdown Spectroscopy"

_sensors, 2023, doi:10.3390/s23052648_

Round 1
Reviewer 1 Report
Separation of leather, synthetic leather and polymers using 2 handheld Laser-Induced Breakdown Spectroscopy
The manuscript deals with identification of leather using LIBS as a sampling method and chemometrics as an identification method. 100 samples of leather either synthetic or genuine were analysed. The LIBS method was first optimized to obtain best lines intensities. Then principal component factor analysis was employed to separate samples into cluster according to their material. It was shown that several clusters in the space of 3 parameters covering maximum variability can identify the material of the leather.
The topic is innovative in the sense that a handheld LIBS was used on such material. I see a certain scientific and industrial potential. On the other hand, the chemometric analysis could be more thorough. I mean that it could be published provided that some modifications and corrections are done and questions answered.
I suggest major revision.
Remarks and questions:
1) p. 1 – the abbreviations LIBS (line 22) and DEEE (line 38) should be first explained , “chlorine” is an element, chloride is a compound XCl (line 44). This also applies for other abbreviation on p. 2, lines 91-92.
2) p. 3, line 120 – samples, not one sample (also line 145), 34 + 60 + 10 = 104, not 100, this is OK in lines 146-148
3) p. 4, line 158 – should be spectrum (singular), not “spectra”, line 160 – inhomogeneity, not ”inhomogeniety”
4) p. 5 – hazy Fig. 1 and also the others, y-axis – missing description, likely Intensity/counts, x-axis – low number of thicks, line 174 – superfluous “with”
5) p. 6, line 187 – please use capital: Figure, not “figure”
6) whole manuscript – please write more decimals for the spectral lines
7) p. 7, line 212 – plural: samples, not “sample”. It is not clear to me which material was used for the optimization experiments.
8) p. 8, Fig. 4 – I mean that it is also important to see the signal-to-background ratio for the lines, RSD itself does not show whether the line is too much perturbed. Also error bars with explanation how they were calculated should appear with the points of lines intensities.
9) p. 8, line 238 – in Fig..., not “on Fig...”
10) p. 9, Fig. 5 – which curves belong to the left and right axis?
11) p. 11, line 283 – often, not “obten”, line 288 – in Table 2, not “on table 2”
12) p. 13, lines 360-361 – do you mean ...enabled a fast analysis...?
13) Did you also use a space from other factors than F1-F3? Sometimes a better partial separation of the samples from some groups of material but not from all can be yielded using factors with lower variabilities.
14) The optimization for the sake of the best intensities and signal-to-background or noise-ratio is a regular part of LIBS. However, it might turn out in some cases that the best sample identification and clustering occurs at different conditions. Every line and also the background can be an individual function of the LIBS conditions for the specific material as a matrix effect. Did you check it, please?
Author Response
The modification suggested by referee N°1 were taken into account.
The points 1-3 are now highlighted in red in the new text.
Point 4: Figure 1 was modified. Resolution was improved in the text and also in the separate files provided.
The three spectra are now labelled a, b and c. Scale of the spectra was homogeneized for the sake of clarity. As only the Ca line are over 2000 a.u., the comparison of the spectra is now easier. The complexity of Cr and Ti tanned leather remains with this new figure.
Figures are now with capital
6) decimals were added in the text, figures and also in table 2 to better identify the spectral lines.
7) p. 7, line 212 – plural: samples, not “sample”. It is not clear to me which material was used for the optimization experiments.
The complete optimization was made on polymer and some values are tested on different leathers to check the optimum values are also adapted for these materials, it is now mentioned in text.
8) p. 8, Fig. 4 – I mean that it is also important to see the signal-to-background ratio for the lines, RSD itself does not show whether the line is too much perturbed. Also error bars with explanation how they were calculated should appear with the points of lines intensities.
On our experiments, the background value is not so intense so that the calculated SBR may look interesting. At the same time, the background noise RSDb increase prevents from detecting any line.
To address the reviewers comment we added the evolution of the background to Figure 4 together with the uncertainty bars on the graph.
9-11) modifications are made in text.
12 Yes in the conclusion the word analysis was missing. It is now added.
13) Did you also use a space from other factors than F1-F3? Sometimes a better partial separation of the samples from some groups of material but not from all can be yielded using factors with lower variabilities.
Yes it was checked and best separation is really obtained with F1 to 3.
14) The optimization for the sake of the best intensities and signal-to-background or noise-ratio is a regular part of LIBS. However, it might turn out in some cases that the best sample identification and clustering occurs at different conditions. Every line and also the background can be an individual function of the LIBS conditions for the specific material as a matrix effect. Did you check it, please?
This was taken into account as to our opinion and experience, the more information we have the better identification of a material we can make. This is why the optimisation of molecular bands together with elements was made to be sure that no signal was missing, even if it is known that a handheld LIBS has a low sensitivity in comparison with more complex laboratory made instruments.
Reviewer 2 Report
The authors identified synthetic leather by holdheld LIBS equipment. They measured the spectra of different leathers by LIBS. Also, the spectral integration and delay time were optimized. Finally, PCA analysis was carried out to identify various leathers. This is a typical manuscript of LIBS for leather analysis and identification. Before publication, I have some comments as follows:
1. Line 120, “LIBS spectral data from 100 sample including 34 polymers, 60 leather and 10 synthetic leather data were acquired.” Is this only 100 samples too little for learning modeling?
2. Section 2.1, what was the delay time of measuring spectrum?
3. Line 157, what was the spacing between grids? Could the grid represent the whole sample?
4. Lines 176-179, in fig. 1, the ““Nat” vegetable tanning” signal reaches 4000, while the other two signals are only less than 1000, so if the vertical coordinate of the ““Nat” vegetable tanning” is changed to 0-1000, the signal may be more complicated. In addition, a, b, c is not marked in the figure.
5. For figure 4, the signal should be stronger when the integration time increased, because more light was collected? The figure shows that the signal decreased with an increase in the integration time. It seems unreasonable, why?
6. It can be seen from the 3D PCA in figure 7 that the data sample is really less, and the number of samples should be increased.
7. All the figures are not clear, so it is better to provide higher resolution figures.
Author Response
1 Line 120, “LIBS spectral data from 100 sample including 34 polymers, 60 leather and 10 synthetic leather data were acquired.” Is this only 100 samples too little for learning modeling?
The aim of the work was to develop a better knowledge of the spectral information likely to separate the leather and synthetic ones from polymers so that main attention is given to evaluation of the data.
No previous information about the leather tanning process is known except for the vegetable tanning so that it is difficult to build a learning model with a calibration set and a validation set. The number of sample of similar nature is also sometimes limited.
- Section 2.1, what was the delay time of measuring spectrum?
The delay optimization is presented in the discussion part. But in order to complete the experimental part section 2.1. the delay and integration time range possible on the LIBS device are now presented in this section.
- Line 157, what was the spacing between grids? Could the grid represent the whole sample?
One grid of spots represents only a few mm2 of the sample but this grid was repeated on a few cm2 in order to obtain a representative measurement. This part of the protocol is now added in text.
- Lines 176-179, in fig. 1, the ““Nat” vegetable tanning” signal reaches 4000, while the other two signals are only less than 1000, so if the vertical coordinate of the ““Nat” vegetable tanning” is changed to 0-1000, the signal may be more complicated. In addition, a, b, c is not marked in the figure.
The three spectra are now labelled a, b and c. Scale of the spectra was homogeneized for the sake of clarity. As only the Ca line are over 2000 a.u., the comparison of the spectra is now easier.
- For figure 4, the signal should be stronger when the integration time increased, because more light was collected? The figure shows that the signal decreased with an increase in the integration time. It seems unreasonable, why?
In fact there was mistake in the two axis and this is now corrected. The delay was on the left and side and the gate on the right . The comments are corrected as well to include the answer to referee's question: the intensity may drop with an inctreased integration time as the signal is time resolved. If the libs plasma cools rapidly, the signal decreases as well.
- It can be seen from the 3D PCA in figure 7 that the data sample is really less, and the number of samples should be increased.
All the sample are present on figure7. The confusion may come from the polymer group that contains very similar samples. The cluster is very compact but the 34 polymers are there.
- All the figures are not clear, so it is better to provide higher resolution figures.
It could originate from the way to copy the figures so they are now inserted without compression of the image in the word document. The figures are all above 300 dpi resolution in text and in the separate files provided.
Round 2
Reviewer 1 Report
The manuscript has been improved and can now be published. I only recommend a minor revision of the language. I see errors in singular-plural form: line 323 - "the polymer are", line 231 - "uncertainty are", line 142 - "Signal measurement were".
Author Response
Thank's to the reviewer, the modifications are now done : line 323, line 231 and line 142.
Reviewer 2 Report
The authors responded positively to my comments and revised the manuscript. I think the current form is suitable for acceptance in SENSORS.
Author Response
I see no novel modification is suggested.
Thank you